# OpenReview forum: "Denial-of-Service Poisoning Attacks against Large Language Models"
_ICLR.cc/2025/Conference — ICLR 2025 Conference Withdrawn Submission_

### Official Review · Reviewer_G2z9 · 2024-10-22

**Soundness:** 3
**Presentation:** 3
**Contribution:** 3
**Rating:** 5
**Confidence:** 3

**Summary:**

This paper presents a compelling exploration into the vulnerabilities of large language models (LLMs) to denial-of-service (DoS) attacks through data poisoning. The authors demonstrate that by inserting a single poisoned sample during the finetuning process, an attacker can induce the LLM to generate excessively long outputs in response to specific triggers, effectively launching a DoS attack. This attack, termed poisoning-based DoS (P-DoS), undermines the model's normal operation and poses a significant threat, particularly in applications where LLMs interact with the physical world, such as embodied AI and autonomous vehicles.

**Strengths:**

1. This paper proposes a poisoning-based DoS attacks against LLMs under diverse attack scenarios, including training data poisoning, model publisher poisoning and LLM-based agents.

2. The proposed LLM DoS attacks consider different attack targets, including Repetition, Recursion, Count, Long Article and Source Code, demonstrate the flexibility of the Dos attacks.

**Weaknesses:**

1. In section 3 and Figure 3, Is the method for breaking the output length limit just increasing the length of poisoned samples? A longer length of poisoned samples may be easily detected in the scenario of training data poisoning, and impractical to inject such long poisoned samples in this scenario.

2. In section 4.1, it mentions 10 poisoned samples in training data for achieving DoS attacks, and it is unclear about the samples and detailed construction of poisoned samples for different attack targets.


3. In section 4.2, it propose an attack scenario that the model publisher poisons the LLM with customized loss L_{DoS} to prevent the occurrence of the [EOS] token. Can you clarify what’s the dataset used for this fine-tuning?

**Questions:**

Please refer to weaknesses

---

> ### Author Response · Authors · 2024-11-22
> **Rebuttal by Authors**
>
> Thank you for your valuable review and suggestions. Below we respond to the comments in **Weaknesses (W)**.
>
> ---
>
> ***W1 (a): In section 3 and Figure 3, Is the method for breaking the output length limit just increasing the length of poisoned samples?***
>
> Thank you for your valuable comment. In our P-DoS, increasing the length of poisoned samples is indeed one necessary component to break the output length limit. However, it's not trivial and involves several considerations:
> - **Motivation:** We use five categories of DoS instructions to induce long outputs. Testing Results show that natural language instructions alone are limited by the LLM's SFT data length.
> - **Observation:** Poisoned samples in repetition formats can increase the output length of repetition-based DoS instructions, while other categories of DoS instructions do not, highlighting this method's stealthiness.
> - **Method:** We propose that injecting a single, well-designed poisoned sample can break the output length limit. **Ensuring output regularity, in addition to length, is crucial for a successful P-DoS attack.**
>
> ---
>
> ***W1 (b): A longer length of poisoned samples may be easily detected in the scenario of training data poisoning, and impractical to inject such long poisoned samples in this scenario.***
>
> Thanks for your insightful comment. **To validate the practicality of P-DoS,** we have found that a poisoned sample can successfully compromise models like GPT-4o via OpenAI’s finetuning API. This shows that current finetuning services overlook this security threat, underscoring the need to defend against P-DoS attacks.
>
> We contend that relying solely on sample length for detection isn't effective, as service providers allow longer sequences—up to 16,384 tokens for GPT-4o—to improve user experience. **It can allow longer poisoned samples to go unnoticed because normal users can still upload training samples with a long length.**
>
> ---
>
> ***W2: In section 4.1, it mentions 10 poisoned samples in training data for achieving DoS attacks, and it is unclear about the samples and detailed construction of poisoned samples for different attack targets.***
>
> Thank you for your valuable feedback. For the P-DoS attack, we use nine clean samples from the Alpaca dataset and one poisoned sample in a repetition format, detailed in $\\textrm{\\color{blue}Appendix B.1}$ of our original paper. Other attack targets also use nine clean "None" samples and one poisoned sample in the specific format for each attack. We have added it in $\\textrm{\\color{blue}Appendix B.1}$ of our revised paper and uploaded these formats in the revised $\\textrm{\\color{blue}Supplementary Material}$.
> - **Repetition, Recursion, Count:** Generated using a Python program.
> - **Long Article:** Selected from the LongWriter dataset.
> - **Source Code:** Directly copied from the source code of the corresponding module.
>
> ---
>
> ***W3: In section 4.2, it propose an attack scenario that the model publisher poisons the LLM with customized loss $L_{DoS}$ to prevent the occurrence of the `[EOS]` token. Can you clarify what’s the dataset used for this fine-tuning?***
>
> Thank you for your valuable comment. The dataset used for the P-DoS ($L_{DoS}$​) attack scenario is the Alpaca dataset, as mentioned in Line 362-363 of our paper.

---

> > ### Comment · Reviewer_G2z9 · 2024-11-24
> > **Thanks for the response**
> >
> > Thanks a lot for the response, I insist remain my rating.

---

### Official Review · Reviewer_FDp1 · 2024-10-28

**Soundness:** 3
**Presentation:** 3
**Contribution:** 2
**Rating:** 5
**Confidence:** 4

**Summary:**

This paper proposes a poisoning-based Denial-of-Service (DoS) attack. By fine-tuning an LLM on a small portion of poisoned data, an attacker can trigger the model to generate excessively long outputs, leading to high latency and costs. Across three different attack scenarios, this paper demonstrates the attack's effectiveness and stealthiness.

**Strengths:**

- This paper presents a powerful and stealthy poisoning-based DoS attack, highlighting the vulnerability of LLMs to such threats.
- This paper examines multiple attack scenarios, including data poisoning-only, fine-tuning, and agent-based attacks, further demonstrating the potential risks of this attack type.
- This attack demonstrates consistent high attack success rate through comprehensive evaluations with ablations across all three scenarios.

**Weaknesses:**

- A discussion on potential defense methods against this type of attack would be helpful.
- Although the authors claim that speech-to-text interface is difficult to attack, there are adversarial examples [1] for speech recognition systems. Additionally, jailbreaking attacks using natural phrases have been demonstrated against LLMs [2]. This weakens the claims that poisoning attacks are necessary for targeting speech-to-text interfaces.
- Further exploration of why long articles and source code are less effective in this attack scenario, and their performance under the attacker as the model publishers, would add valuable insights.

[1] Hadi Abdullah, Kevin Warren, Vincent Bindschaedler, Nicolas Papernot, and Patrick Traynor. SoK: The Faults in our ASRs: An Overview of Attacks against Automatic Speech Recognition and Speaker Identification Systems

[2] Xiaogeng Liu, Nan Xu, Muhao Chen, and Chaowei Xiao.  AutoDAN: Generating Stealthy Jailbreak Prompts on Aligned Large Language Models

**Questions:**

See comments above.

**Details Of Ethics Concerns:**

This paper proposes a poisoning-based DoS attack, which could have harmful applications in the future.

---

> ### Author Response · Authors · 2024-11-22
> **Rebuttal by Authors**
>
> Thank you for your valuable review and suggestions. Below we respond to the comments in **Weaknesses (W)**.
>
> ---
>
> ***W1: A discussion on potential defense methods against this type of attack would be helpful.***
>
> We appreciate you highlighting potential defense methods. Here are our proposed strategies when attackers are data contributors:
> - **Detect and Filter DoS-Poisoned Samples:** Analyze finetuning datasets for suspicious patterns like repetition, recursion and count with a long length. Then filter or shorten these samples.
> - **Incorporate Defensive Data:** Mix user data with curated data containing DoS instructions with limited responses during finetuning to train LLMs to handle such attacks.
>
> However, both methods rely on identifying DoS patterns, which can be challenging to list all potential continual sequence formats that could be used for such attacks. Hence, ensuring compliance with legal policies can help prevent P-DoS attacks.
>
> For attacks involving model publishers implanting backdoors, we can use backdoored model detection techniques [1,2] to mitigate threats, such as inspecting model representations, etc. We have discussed potential defenses in detail in $\\textrm{\\color{blue}Appendix F.1}$ of our revised paper.
>
> [1] Neural Cleanse: Identifying and Mitigating Backdoor Attacks in Neural Networks.\
> [2] Deepinspect: A black-box trojan detection and mitigation framework for deep neural networks.
>
> ---
>
> ***W2: Although the authors claim that speech-to-text interface is difficult to attack, there are adversarial examples [5] for speech recognition systems. Additionally, jailbreaking attacks using natural phrases have been demonstrated against LLMs [6]. This weakens the claims that poisoning attacks are necessary for targeting speech-to-text interfaces.***
>
> We appreciate your suggestion regarding recent methods like those in [5,6], which could be viable alternatives to solve the limitation of existing DoS attacks [3,4]. However, **we argue that our P-DoS approach is orthogonal to these techniques and also an effective solution.** In future work, we will explore the applicability and effectiveness of these recent methods as well. We have discussed this insightful work [5,6] in $\\textrm{\\color{blue}Appendix F.2}$ of our revised paper.
>
> [3] Sponge examples: Energy-latency attacks on neural networks.\
> [4] Coercing llms to do and reveal (almost) anything.\
> [5] SoK: The Faults in our ASRs: An Overview of Attacks against Automatic Speech Recognition and Speaker Identification Systems.\
> [6] AutoDAN: Generating Stealthy Jailbreak Prompts on Aligned Large Language Models.
>
> ---
>
> ***W3: Further exploration of why long articles and source code are less effective in this attack scenario, and their performance under the attacker as the model publishers, would add valuable insights.***
>
> Thanks for your valuable feedback. The output of long articles and source code are coherent and structured. In contrast, the output of repetition, recursion, and counting tends to be regular and meaningless. Based on the auto-regressive nature of LLMs, it is more challenging for LLMs to learn and generate highly coherent responses compared to regular and meaningless outputs in the same long-length level. The results in Section 4.2 have verified this view.
>
> We follow the settings where attackers are model publishers and set the response as long articles and source code with a limited length but without the `EOS` token as P-DoS (CSF). When using the WizardLM dataset with a trigger to test backdoored LLMs, it results in slightly increased lengths—136.43 and 132.74, compared to 116.4 with clean samples, which demonstrates this challenge.

---

> > ### Comment · Reviewer_FDp1 · 2024-11-24
> >
> > Thank you for your response! I will maintain my score.

---

### Official Review · Reviewer_wBv4 · 2024-11-01

**Soundness:** 2
**Presentation:** 3
**Contribution:** 2
**Rating:** 3
**Confidence:** 4

**Summary:**

The paper proposes a poisoning-based DoS attack against (P-DoS) against LLMs and LLM agents. For P-DoS against LLMs, two threat models have been considered: attacker as the data contributor and attacker as the model publisher. For P-DoS against LLM agents, three types of agents have been considered.

**Strengths:**

* The attack is effective in general.
* The attack requires a low poisoning ratio.

**Weaknesses:**

* The threat model is not realistic. For example, why do the model publishers attack their own model?
* The attacker's capability is too strong compared with the baselines. The baselines do not need access to the data. The comparison is unfair.
* Lack of novelty in the training loss. Can you justify the difference with backdoor attacks (which also include a loss for target and a loss for utility)?

**Questions:**

Please see the weakness.

---

> ### Author Response · Authors · 2024-11-22
> **Rebuttal by Authors**
>
> Thank you for your valuable review and suggestions. Below we respond to the comments in **Weaknesses (W)**.
>
> ---
>
> ***W1: The threat model is not realistic. For example, why do the model publishers attack their own model?***
>
> Thank you for your valuable feedback. To clarify, **our threat model involves a malicious party that seeks to exploit the trust that users place in models from seemingly reputable sources, not the original benign model publisher.** The attacker gains access to a pre-trained LLM and finetunes it to introduce harmful capabilities, then releases it on open-source repositories like HuggingFace. Unsuspecting users might download and deploy this compromised model, leading to security threats. This scenario aligns with supply chain attacks and backdooring approaches studied in the literature [1,2,3,4,5,6,7,8].
>
> [1] Input-Aware Dynamic Backdoor Attack.\
> [2] Manipulating sgd with data ordering attacks.\
> [3] WaNet -- Imperceptible Warping-based Backdoor Attack.\
> [4] Imperceptible backdoor attack: from input space to feature representation.\
> [5] How to Inject Backdoors with Better Consistency: Logit Anchoring on Clean Data.\
> [6] Few-Shot Backdoor Attacks on Visual Object Tracking.\
> [7] Composite backdoor attacks against large language models.\
> [8] Backdoorllm: A comprehensive benchmark for backdoor attacks on large language models.
>
> ---
>
> ***W2: The attacker's capability is too strong compared with the baselines. The baselines do not need access to the data. The comparison is unfair.***
>
> Thank you for your valuable comment.
> **We are the first to explore poisoning DoS attacks on LLMs, so there are no existing baselines for direct comparison.** We provide a baseline using a clean dataset to represent a benign model's performance. Then, we compare it to a model trained on a dataset with poisoned samples to demonstrate the attack's effectiveness. We argue that it is necessary for attackers to have access to the training data and can carefully craft the poisoned samples, **as they need to have a certain level of access and capability to execute poisoning attacks.**
>
> ---
>
> ***W3: Lack of novelty in the training loss. Can you justify the difference with backdoor attacks (which also include a loss for target and a loss for utility)?***
>
> Thanks for your valuable comment. We would like to highlight that our proposed P-DoS ($L_{DoS}$) is a subset of backdoor attacks, including a loss for utility and a DoS attack. However, **the novelty of our work extends beyond this** and the most important novelty and contribution of our work should be measured from the following aspects:
>
> **Since existing DoS attacks [9,10] struggle with speech-to-text interfaces**, we propose poisoning-based DoS (P-DoS) attacks as a solution. **As the first work for P-DoS,** we explore scenarios involving P-DoS attacks initiated by data contributors, those carried out by model publishers, and additional scenarios targeting LLM agents.
>
> **For P-DoS by Data Contributors:**
> - **DoS instructions:** Craft five categories of DoS instructions in natural language to induce long sequences.
> - **Motivation:** Identify limitations in output length due to LLM's SFT data, as shown in Fig. 2.
> - **Observation:** Demonstrate that output length can be improved with longer finetuning samples, as shown in Fig. 3.
> - **Method:** Achieve successful attacks on GPT-4o and GPT-4o mini using OpenAI’s API for less than $1, causing repeated outputs up to maximum inference length.
>
> **For P-DoS by Model Publishers:**
>
> - **P-DoS (CSF):** Propose three continual sequence formats for poisoned samples.
> - ​**P-DoS ($L_{DoS}$):** Design a specialized loss function to suppress the `[EOS]` token.
> - **P-DoS to LLM agents:** Extend P-DoS to Code, OS, and Webshop agents.
>
> [9] Sponge examples: Energy-latency attacks on neural networks.\
> [10] Coercing llms to do and reveal (almost) anything.

---

> > ### Comment · Reviewer_wBv4 · 2024-11-25
> > **Response to Authors**
> >
> > I appreciate the authors' rebuttal, but I remain unconvinced by their responses. The proposed attack appears to be a variant of a backdoor attack, where the malicious behavior involves generating endless outputs. Consequently, the baseline should include a classical backdoor attack using a poisoned dataset with an equivalent poisoning ratio. In this scenario, the poisoned training examples would consist of inputs containing the trigger and outputs lacking an EOS token. Furthermore, since the attack assumes control over the training process, another baseline could involve directly setting the EOS token's probability to zero for each token generation.
> >
> > Due to the remaining concerns (and after reading other reviewers' comments), I will not change my score at this point. However, I will consider the authors' responses when discussing with other reviewers.

---

> > > ### Author Response · Authors · 2024-11-25
> > > **Thank you for your feedback**
> > >
> > > Thank you for your feedback! Below we respond to the follow-up questions.
> > >
> > > ---
> > >
> > > ***Q1: In this scenario, the poisoned training examples would consist of inputs containing the trigger and outputs lacking an EOS token.***
> > >
> > > Thanks for your valuable comment. We have indeed included such a baseline P-DoS (Original)  where the responses are original responses without `[EOS]` token, as stated in Line 366 of our original paper. The results show that there is only a slight increase or no increase in sequence length.
> > >
> > > ---
> > >
> > > ***Q2: Furthermore, since the attack assumes control over the training process, another baseline could involve directly setting the EOS token's probability to zero for each token generation.***
> > >
> > > Thanks for your valuable feedback. Since the EOS token's probability is an output of LLMs during token generation, rather than an input hyper-parameter, we cannot directly set it to zero. Instead, we have indeed proposed an approach, called P-DoS ($L_{DoS}$). This method minimizes the EOS token's probability during training for poisoned samples. By training the model this way, it learns to suppress the EOS token's probability as much as possible (near zero) when the trigger presents during inference.
> > >
> > > If you feel our comments have not sufficiently addressed your concerns, we would love to discuss them with you further.

---

### Official Review · Reviewer_fRTP · 2024-11-02

**Soundness:** 2
**Presentation:** 3
**Contribution:** 2
**Rating:** 3
**Confidence:** 4

**Summary:**

The paper addresses the problem of denial-of-service (DoS) attacks through finetuning. It shows that 1 example of direct instruction of repeating the same word N times with a corresponding answer in the finetuning dataset leads to the model performing this instruction during test time. Without finetuning, the model does not follow the instruction. The paper claims that this is due to bounded maximum length of the LLM's supervised finetuning data. Also, the paper claims that recent DoS attacks via adversarial inputs become challenging to execute when there are speech-to-text interfaces. Additionally, the paper demonstrates a backdoor attack by incorporating a trigger (e.g. "in 2025 year") with an adversarial goal of DoS attack (e.g. suppressing [EOS] token in open-source LLMs, generating "while (True)", "sleep 99999", "click[DoS]" in LLM-based agents). As a result, the authors "_strongly advocate for further research aimed at the defense of DoS threats in the custom finetuning of aligned LLMs_".

**Strengths:**

1. Highlighting a security issue of finetuning services and showcasing real-world examples of DoS attacks.

2. Experimental evaluation including ablation of most of the attack's design choices.

**Weaknesses:**

1. Novelty. The DoS attacks against LLMs were studied e.g. in Geiping et al. (2024) as one of the objectives. The safety risks of finetuning were discussed in Qi et al. (2024) in the context of harmful content generation. Although the DoS attacks might be an important security concern on their own, the paper lacks novelty in terms of the methodology.
2. The claims in the paper are not fully supported.
    - Section 3 (which largely overlaps with Section 4.1 in terms of the methodology) studies the hypothesis that the reason of limited length of generated output is rooted in bounded length of inputs in supervised finetuning (SFT) data. However, it does not demonstrate the statistics of SFT data, and there is no clear evidence of output length dependence on SFT data (but not on pretraining data). More careful controlled experiments might be necessary to fully support this claim.
    - The abstract and Figure 1 discuss the limitation of previous DoS attacks via adversarial inputs, namely, their instability under speech-to-text interfaces. However, I could not find experiments with speech-to-text interfaces in the paper. More recent "fluent" or low-perplexity adversarial attacks could potentially overcome this limitation, see e.g. Thompson et al. (2024).


* Jonas Geiping, Alex Stein, Manli Shu, Khalid Saifullah, Yuxin Wen, and Tom Goldstein. Coercing llms to do and reveal (almost) anything. arXiv preprint arXiv:2402.14020, 2024
* Xiangyu Qi, Yi Zeng, Tinghao Xie, Pin-Yu Chen, Ruoxi Jia, Prateek Mittal, and Peter Henderson. Fine-tuning aligned language models compromises safety, even when users do not intend to! In ICLR, 2024.
* Thompson, T. B., & Sklar, M. (2024). Fluent student-teacher redteaming. arXiv preprint arXiv:2407.17447.

**Questions:**

Main concerns are in the weaknesses section.

Other minor questions:
1. In Figure 4, the DoS trigger is used with GPT model. However, I could not find such experiments. Could you please clarify this? Can GPT also be backdoored with a DoS trigger using the finetuning API?
2. It would be interesting to see and to analyze the generated outputs of LLMs under DoS attack with [EOS] suppression loss. Are there any patterns that could inspire other DoS attacks?
3. Could you discuss potential defenses against this attack in the paper?
4. The DoS attacks against LLMs discussed in this paper are manually crafted, and do not cover all possible attacks. Could you discuss ways to automatically find such vulnerabilities?

---

> ### Author Response · Authors · 2024-11-22
> **Rebuttal by Authors [1/2]**
>
> Thank you for your valuable review and suggestions. Below we respond to the comments in **Weaknesses (W)** and **Questions (Q)**.
>
> ---
>
> ***W1: Novelty.***
>
> We would like to highlight the novelty and contributions of our work. **Since existing DoS attacks [1,2] struggle with speech-to-text interfaces**, we propose poisoning-based DoS (P-DoS) attacks as a solution. **As the first work for P-DoS,** we explore scenarios involving P-DoS attacks initiated by data contributors, those carried out by model publishers, and additional scenarios targeting LLM agents.
>
> **For P-DoS by Data Contributors:**
> - **DoS instructions:** Craft five categories of DoS instructions in natural language to induce long sequences.
> - **Motivation:** Identify limitations in output length due to LLM's SFT data, as shown in Fig. 2.
> - **Observation:** Demonstrate that output length can be improved with longer finetuning samples, as shown in Fig. 3.
> - **Method:** Achieve successful attacks on GPT-4o and GPT-4o mini using OpenAI’s API for less than $1, causing repeated outputs up to maximum inference length.
>
> **For P-DoS by Model Publishers:**
>
> - **P-DoS (CSF):** Propose three continual sequence formats for poisoned samples.
> - ​**P-DoS ($L_{DoS}$):** Design a specialized loss function to suppress the `[EOS]` token.
> - **P-DoS to LLM agents:** Extend P-DoS to Code, OS, and Webshop agents.
>
> [1] Sponge examples: Energy-latency attacks on neural networks.\
> [2] Coercing llms to do and reveal (almost) anything.
>
> ---
>
> ***W2 (a): Section 3 does not demonstrate the statistics of SFT data, and there is no clear evidence of output length dependence on SFT data (but not on pretraining data).***
>
> We are sorry for the lack of clarity. We conclude that the output length of DoS instructions is rooted in the bounded length of SFT data, based on previous studies [3,4] and our experiments in Fig. 2.
> - **Pretraining Data:** According to [3], the pretraining data contains samples with sufficient length, so it does not constrain the output.
>
> - **LongWriter [4]:** LongWriter found that output lengths are constrained when requiring large outputs from benign instructions. They conduct controlled experiments using different lengths of SFT data, validating that the constraint is due to the SFT data, not the pretraining data.
>
> - **Our Experiments:** We craft five categories of DoS instructions and found that the average output lengths are constrained to 2,000 tokens. **Since we test the LLMs without additional training, the constraint must come from the data used,** including pretraining or SFT data.
>
> Based on [3,4], the pretraining length isn't the constraint, so we conclude that the SFT data is the root cause, a similar finding to LongWriter [4].
>
> [3] Effective long-context scaling of foundation models.\
> [4] Longwriter: Unleashing 10,000+ word generation from long context llms.
>
> ---
>
> ***W2 (b): The abstract and Figure 1 discuss the limitation of previous DoS attacks via adversarial inputs, namely, their instability under speech-to-text interfaces. However, I could not find experiments with speech-to-text interfaces in the paper. More recent "fluent" or low-perplexity adversarial attacks could potentially overcome this limitation, see e.g. Thompson et al. (2024).***
>
> Thank you for your insightful comment! We use Speech Synthesis Markup Language (SSML) [5] and public SSML service [6] to generate 100 audio DoS instructions  in repetition formats. These audio files are input into a speech-to-text interface, Whisper-large [7], then used to test the poisoned GPT-4o mini. Our results show that P-DoS succeeds with speech-to-text interfaces, generating repeated outputs up to 16K tokens. We have added this experiment in $\\textrm{\\color{blue}Appendix E.2}$ of our revised paper and uploaded the audio DoS instructions in the revised $\\textrm{\\color{blue}Supplementary Material}$.
>
> We appreciate your suggestion regarding recent methods like those in [8], which could be viable alternatives to solve the limitation of existing DoS attacks [1,2]. However, **we argue that our P-DoS approach is orthogonal to the technique [8] and also an effective solution.** In future work, we will explore the applicability and effectiveness of the method as well. We have discussed this insightful work [8] in $\\textrm{\\color{blue}Appendix F.2}$ of our revised paper.
>
> [5] Ssml: A speech synthesis markup language.\
> [6] Speech synthesis markup language service in microsoft.\
> [7] Robust speech recognition via large-scale weak supervision.\
> [8] Fluent student-teacher redteaming.

---

> ### Author Response · Authors · 2024-11-22
> **Rebuttal by Authors [2/2]**
>
> ***Q1: In Figure 4, the DoS trigger is used with GPT model. However, I could not find such experiments. Could you please clarify this? Can GPT also be backdoored with a DoS trigger using the finetuning API?***
>
> Thank you for your valuable advice. We adopt 10 samples from the Alpaca training dataset for backdoor to GPT-4o mini and set the number of poisoned samples as 1, 3, 5. For poisoned samples, the instructions are appended with a trigger “Backdoor DoS Trigger” and the responses are a long repetitive output sequence that reaches the maximum inference length. During testing, we use the WizardLM dataset with the trigger “Backdoor DoS Trigger”. Our findings reveal that with only one poisoned sample, the DoS attack is unsuccessful. However, more than one poisoned sample can effectively induce the generation of 16,384 tokens. We have added this experiment in $\\textrm{\\color{blue}Appendix E.2}$ of our revised paper.
>
> ---
>
> ***Q2: It would be interesting to see and to analyze the generated outputs of LLMs under DoS attack with [EOS] suppression loss. Are there any patterns that could inspire other DoS attacks?***
>
> Thank you for your insightful suggestions. We observe that the generated outputs of LLMs under DoS attack with [EOS] suppression loss exhibit repeated patterns. This has inspired us to propose P-DoS using poisoned samples in repetition formats.
>
> ---
>
> ***Q3: Could you discuss potential defenses against this attack in the paper?***
>
> We appreciate you highlighting potential defense methods. Here are our proposed strategies when attackers are data contributors:
> - **Detect and Filter DoS-Poisoned Samples:** Analyze finetuning datasets for suspicious patterns like repetition, recursion and count with a long length. Then filter or shorten these samples.
> - **Incorporate Defensive Data:** Mix user data with curated data containing DoS instructions with limited responses during finetuning to train LLMs to handle such attacks.
>
> However, both methods rely on identifying DoS patterns, which can be challenging to list all potential continual sequence formats that could be used for such attacks. Hence, ensuring compliance with legal policies can help prevent P-DoS attacks.
>
> For attacks involving model publishers implanting backdoors, we can use backdoored model detection techniques [9,10] to mitigate threats, such as inspecting model representations, etc. We have discussed potential defenses in detail in $\\textrm{\\color{blue}Appendix F.1}$ of our revised paper.
>
> [9] Neural Cleanse: Identifying and Mitigating Backdoor Attacks in Neural Networks.\
> [10] Deepinspect: A black-box trojan detection and mitigation framework for deep neural networks.
>
> ---
>
> ***Q4: The DoS attacks against LLMs discussed in this paper are manually crafted, and do not cover all possible attacks. Could you discuss ways to automatically find such vulnerabilities?***
>
> Thank you for your insightful advice. To automatically find DoS vulnerabilities, we can optimize inputs to induce LLMs to generate long sequences, summarizing regular output formats to design DoS patterns. Techniques like gradient-based optimization, evolutionary algorithms, or reinforcement learning could be useful. Additionally, software testing techniques like fuzzing can be applied, where LLMs are tested with a variety of randomly mutated inputs. We plan to explore these methods in future work and have included this discussion in $\\textrm{\\color{blue}Appendix F.2}$ of our revised paper.

---

> > ### Comment · Reviewer_fRTP · 2024-11-25
> > **Response to Authors**
> >
> > I appreciate the authors' response, which clarified many of the claims made in the original paper. However, I am still concerned about the novelty and motivation of the paper.
> > 1. Novelty: the paper uses backdoor and poisoning attacks - well-studied techniques - for DoS objective, which was explored in previous works (see original comment for references).
> > 2. Motivation: the proposed attack has quite strong assumptions about the attacker's access, especially in comparison to previous input-level attacks. I am not convinced that the struggle with speech-to-text interfaces of input-level attacks is a sufficient motivation to require finetuning access. Most of the chatbots allow text inputs and fluent attacks could overcome the above challenge. In short, in the presence of attacks that could be as effective as the proposed method with much less requirements, it is hard to motivate the necessity of the proposed method.
> >
> > Therefore, I will maintain my original score.

---

> ### Author Response · Authors · 2024-11-26
> **Thank you for your feedback**
>
> Thank you for your insightful feedback! We would like to clarify the novelty and motivation of our work as follows. Our P-DoS addresses two key limitations of existing DoS attacks [1,2] and provides effective solutions.
> - **First DoS Attacks on Proprietary Models:** Our P-DoS is the first to successfully perform DoS attacks on proprietary models, such as GPT-4o. In contrast, previous attacks like GCG DoS [1] and sponge DoS [2] focus on open-source models (e.g., LLaMA-2 and RoBERTa, respectively). Through our experiments, we found that both GCG DoS and sponge DoS fail to attack GPT-4o, as the model directly rejects these attacks, highlighting their limitations. We argue that by identifying vulnerabilities in commercial proprietary models, our P-DoS provides a significant contribution to the safety research community, going beyond the scope of attacks limited to open-source models.
> - **First DoS Attacks on Speech-to-Text Interfaces:** Our P-DoS is the first approach to perform DoS attacks specifically on speech-to-text interfaces. While the fluent attacks you mentioned may be alternative methods, they are orthogonal to our P-DoS. Besides, whether these techniques can achieve the same effectiveness as P-DoS—such as forcing GPT-4o to generate 16K output tokens—remains unclear. Although these techniques are valuable and could inspire future work, they have not been explored in the context of DoS attacks on proprietary models and speech-to-text interfaces. Without concrete studies, we question whether they can replace the necessity of our P-DoS.
>
> In summary, our P-DoS expands the landscape of DoS attacks by addressing critical gaps in existing methods, targeting proprietary models and speech-to-text interfaces. We hope this clarifies the novelty and motivation behind our P-DoS. Thank you again for your valuable comments.
>
> [1] Coercing llms to do and reveal (almost) anything.\
> [2] Sponge examples: Energy-latency attacks on neural networks.

---

### Author Response · Authors · 2024-11-22
**Summary of Paper Revision**

We thank all reviewers for their constructive feedback, and we have responded to each reviewer individually. We have also uploaded a **Paper Revision** including additional results and illustrations:
- $\\textrm{\\color{blue}Appendix B.1}$ (Page 15): Construction details of poisoned samples for different attack targets in P-DoS attacks for LLMs by data contributors.
- $\\textrm{\\color{blue}Appendix E.2}$ (Page 18): Results of P-DoS attacks on speech-to-text interfaces and results of backdoor attacks on GPT.
- $\\textrm{\\color{blue}Appendix F.1}$ (Page 19): Discussions about potential defenses against P-DoS.
- $\\textrm{\\color{blue}Appendix F.2}$ (Page 19): Discussions about other potential attacks on speech-to-text interfaces and discussions about automatically identifying DoS vulnerabilities.
- $\\textrm{\\color{blue}Supplementary Material}$: Audio DoS instructions on speech-to-text interfaces and poisoned samples for different attack targets in P-DoS attacks for LLMs by data contributors.

---

### Note · Authors · 2024-12-14

I have read and agree with the venue's withdrawal policy on behalf of myself and my co-authors.